# Reasons for Increased Caesarean Section Rate in Vietnam: A Qualitative Study among Vietnamese Mothers and Health Care Professionals

**DOI:** 10.3390/healthcare8010041

**Published:** 2020-02-21

**Authors:** Mizuki Takegata, Chris Smith, Hien Anh Thi Nguyen, Hai Huynh Thi, Trang Nguyen Thi Minh, Louise Tina Day, Toshinori Kitamura, Michiko Toizumi, Duc Anh Dang, Lay-Myint Yoshida

**Affiliations:** 1Department of Pediatric Infectious Diseases, Institute of Tropical Medicine, Nagasaki University, Nagasaki 852-8523, Japan; toizumi@nagasaki-u.ac.jp (M.T.); lmyoshi@nagasaki-u.ac.jp (L.-M.Y.); 2Maternal, Adolescent, Reproductive & Child Health (MARCH), London School of Hygiene & Tropical Medicine, London WC1E 7HT, UK; christopher.smith@lshtm.ac.uk (C.S.); louise-tina.day@lshtm.ac.uk (L.T.D.); 3Graduate School of Tropical Medicine & Global Health, Nagasaki University, Nagasaki 852-8523, Japan; 4National Institute of Hygiene and Epidemiology, Hanoi 100000, Vietnam; hienanh75@yahoo.com (H.A.T.N.); dangducanh.nihe@gmail.com (D.A.D.); 5Collaborative Office for Public Health Research, Khanh Hoa Health Service Governmental Office, Nha Trang 650000, Vietnam; huynhthihai46@gmail.com (H.H.T.); khoinguyen08112014@gmail.com (T.N.T.M.); 6Kitamura Institute of Mental Health Tokyo, Tokyo 1510063, Japan; kitamura@institute-of-mental-health.jp; 7Department of Psychiatry, Graduate School of Medicine, Nagoya University, Nagoya 4668550, Japan

**Keywords:** caesarean section, determinants, health care professionals, pregnant women, qualitative interview, Vietnam

## Abstract

The Caesarean section rate in urban Vietnam is 43% in 2014, which is more than twice the recommended rate (10%–15%) by the World Health Organization. This qualitative study aims to identify the perceptions of pregnant mothers and health care professionals on the medical and social factors related to the increased Caesarean section rate in Vietnam. A qualitative descriptive study was conducted among pregnant mothers and healthcare professionals at two public hospitals in Nha Trang city. A content analysis was adopted in order to identify social and medical factors. As a result, 29 pregnant women and 19 health care professionals were invited to participate in the qualitative interviews. Private interviews were conducted with 10 women who wished to have a Caesarean section, and the others participated in focus group interviews. The main themes of the social factors were ‘request for Caesarean section,’ ‘mental strain of obstetricians,’ and ‘decision-making process.’ To conclude, this qualitative study suggests that there were unnecessary caesarean sections without a clear medical indication, which were requested by women and family members. Psychological fear occurred among women and family, and doctors were the main determinants for driving the requests for Caesarean section, which implies that education and emotional encouragement is necessary by midwives. In addition, a multi-faced approach including a mandatory reporting system in clinical fields and involving family members in antenatal education is important.

## 1. Introduction

The concept of too little, too late (TLTL) and too much, too soon (TMTS) was first introduced by Miller et al. [1], describing two groups of medical problems in the world. TLTL is defined as ‘insufficient evidenced-based care—inadequate access to services, resources, or evidence-based care,’ which occurs in low-income countries (LICs) [1]. On the other hand, TMTS refers to the ‘over-medicalization of normal antenatal, intrapartum, and postnatal care’, which is mainly seen in middle-income countries (MICs) [1]. TMTS is an emergent concern of over-medicalization leading to subsequent harm associated with medical interventions [1,2].

A Caesarean section (CS) is a life-saving mode of delivery necessary for mother and baby, however, CS has been recognized as a major example representing TMTS [1]. The CS rate is rising in many countries worldwide, exceed the rate range of 10% to 15% recommended by the WHO [2,3,4]. Countries reporting a rate of CS over 40% are in Latin America, the Caribbean, and China [4]. The CS rate of Vietnam has rapidly increased from 10% in 2002 to 28% in 2014 [5], and the urban rate is very high especially (2014: 43%) [5]. According to the recent study conducted in Da Nang city [6], located in an urban area in central Vietnam, the overall CS rate was 58.6%. 

Both medical and social factors related to rising CS were reported in previous studies from other Asian countries such as previous CS birth [7], a request for CS without medical indication [8], increased obesity [7], higher socio-economic status [9,10,11,12,13], private hospitals [14], pregnancies with a male fetus [14], having access to full-insurance payment [14], and pregnancies after infertility treatment [8]. However, there is limited information on social and medical determinants that drives the increased CS rate in Vietnam. To the best of our knowledge, there are only two studies that explored determinants for the CS rate in Vietnam. Hoa et al. (2012) found that mothers of a boy child were more likely to receive CS surgery than those giving birth to a girl in a province of northern Vietnam [15]. Giang et al. (2018) also identified that the CS rate of private hospitals in Da Nang city was 70.6%, and correlated with older aged woman (≥30 years), having a history of abortion, an office worker, having a boy child, and with higher neonatal birth weight [6]. 

In order to understand more about determinants for the increased CS rate in Vietnam, a qualitative exploration is necessary as a first step for planning a large-sized quantitative survey in the near future. Specifically, the birth culture of Vietnam, including the perception of the mode of delivery among women [15,16] and a woman’s autonomy [17,18] may be different from other countries. In China and Taiwan, CS has been regarded as a safe option among some of the Chinese mothers, who also use traditional Chinese astrology to decide whether CS; this could be one of the reasons for the rise of the CS rate [10,14]. Therefore, it important to understand the deep-rooted social, psychological, and cultural aspects lying behind the rising CS rate. 

The objectives of this qualitative study were to identify the perceptions of pregnant mothers and health care professionals for medical and social factors related to the increased CS rate in Vietnam. 

## 2. Materials and Methods

### 2.1. Design and Participants

A qualitative, descriptive design was selected. For health care professionals and pregnant mothers who do not wish to have a CS at the time of recruitment, mini-sized focus group interviews with a maximum number of five participants per session were conducted in order to ensure enough time to discuss the topic in a relaxing atmosphere [19]. The focus group interviews were conducted separately, and separated into each group by their backgrounds; obstetricians, midwives, and pregnant mothers. On the other hand, for the pregnant women who wish to, or have chosen to have a CS without having any medical reason, we conducted an interview privately because these women might feel uncomfortable expressing their own insights to others. 

Two settings were chosen for recruiting participants. Setting A: Pregnant women were invited to participate in an interview at A hospital which is a public facility, which mainly conducts antenatal checkups and assists 200 normal deliveries in the city. Setting B: Health care professionals were invited to participate in this study at B tertiary hospital, a public facility assisting more than 4000 deliveries annually in Nha Trang. 

Exclusion criteria for pregnant women were: (i) <20 years of age; (ii) unable to communicate in or read Vietnamese; (iii) having a planned CS birth due to medical reasons; (iv) < 12 gestational weeks, or (v) having serious pregnancy complications. An adolescent is defined as a person aged between 10 and 19 according to WHO [20], who is not fully matured, has three ethical issues; inability to legally consent, issues of confidentiality, and decision-making abilities [21]. Although many previous studies regard the age of 18 as being capable of these ethical issues, the authors decided to exclude any person who is under the age of 20 years following the WHO definition. As for health care professionals, obstetricians and midwives, those working at B hospital were recruited but those who have worked less than one year were excluded from this study.

With regard to analysis, a content analysis was adopted. Content analysis is often used as a combination of qualitative and quantitative methods. The qualitative way draws the presence, meanings, and relationships of certain words, themes, or concepts within some qualitative data. The quantitative way focuses on counting the occurrence of certain words and phrases in different contexts [19].

### 2.2. Procedure

This study was conducted between April and July 2018. Eligible pregnant women were directly recruited at the time of their antenatal health check-up by midwives working at A hospital. Eligible health care professionals of B hospital were directly invited to participate in this study by two staff working at the Khanh Hoa Health Service Governmental Office. The eligible candidates were verbally told about the purposes of the study prior to the interview and given an informed consent form. If a candidate agreed for participation, he or she was asked to sign the informed consent form on the interview date. 

There were three trained Vietnamese interviewers, including two obstetricians and one midwife. Each interviewer facilitated interviews in a private room at the two hospitals. An interview guide was separately developed for pregnant women and health care professionals (Appendix A). Pregnant women were asked the reason for requesting a CS and factors that may be related to increased CS, e.g., fear of childbirth, and ideal care to reduce the CS rate. In addition to these questions, a woman who participated in a private interview was asked the reason why she wished (or requested) CS. Health care professionals were asked to answer medical, social and psychological reasons for increased the CS rate in Nha Trang city, and what the ideal care was to reduce the CS rate (Appendix A). After obtaining the participants’ permission, the focus group interviews and the private interviews were audio-recorded.

The audio records were transcribed by translators from the Khanh Hoa Health Service Governmental Office, then translated into English by three Vietnamese bilinguals. Consistencies between the Vietnamese transcription and English version were carefully cross-checked by the authors H.T.N. and H.T.H. Following a method of content analysis [19], first, relevant statements were extracted, coded by unit, subcategorized, and categorized by MT and CS. Next, researchers (TK and LTD) independently applied categorizations to each unit statement in order to calculate the agreement rates.

## 3. Results

Twenty-nine pregnant women and nineteen health care professionals (four medical doctors, fifteen midwives) participated in the study. Ten pregnant women who requested a CS birth were asked to participate in a private interview in order to guarantee their privacy, whereas the others were enrolled in focus group interviews. Each session lasted from a minimum of 40 to a maximum of 60 minutes.

Table 1 displays the characteristics of the participants. Out of 29 pregnant women, seventeen were primiparas. Most of the participants had no serious pregnancy complexity but one participant had a minor complication (anemia). Their ages varied from 20 to 38 years old. Five midwives were between 25 and 29 years of age, however, a majority of healthcare professionals were over 35 years of age and had more than ten years of working experience.

### 3.1. Social Factors

Table 2 and Table 3 shows social and medical factors related to the increased CS in this region. In terms of social factors (Table 2), three themes emerged; (1) request for CS; (2) mental strain of obstetricians and; (3) decision-making process.

#### 3.1.1. Theme 1: Request for CS

Nine categories and thirty-two subcategories emerged. Categories were grouped into two; psychological aspects of women and their family members (Categories 1–6), and the community aspects such as cultural belief, geographic, and institutional reasons (Categories 7–9). 

First, with regard to the psychological aspects, *fear of vaginal childbirth* was the main reason for requesting a CS among the pregnant women who participated in the private interview, however, the subjects of fear these women were expressing varied: fear of labor pain was most frequently expressed by all pregnant mothers of the private interview group and some of whom participated in the group interview. The unexpected risk for mother and child, such as massive bleeding, fetal distress, and obstructed labor was the second reason that most of the participants expressed. In addition, some of the participants of the private interview group were afraid of episiotomy and suffering for a long period of time during labor. All of the multiparous women at the private interview group mentioned that they chose CS due to their negative experience from their previous childbirth, which was described as ‘painful’ and ‘isolated.’ A few participants that requested a CS were afraid of injection, losing control such as becoming panicky in front of family and/or health care professionals. One pregnant woman from the private interview group and three health care professionals reported forceps or vacuum instrument sometimes being recognized as a threat which would damage the unborn child’s brain as a concern among some of the parents. 

In addition, *fear of vaginal childbirth* was often accompanied by *concern about older age*, and/or *lack of self-efficacy for childbirth*. Some of the pregnant women requested a CS because they were not confident about giving vaginal birth due to lack of energy, or concerned higher risk of delivery complications due to older age. Even younger pregnant mothers expressed they were not confident about giving vaginal birth. 

Furthermore, it was found that some of the participants who wished for a CS had a strong *belief that CS has a social advantage or/and that CS is safe and more comfortable.* For those who requested CS, CS is regarded as being safer, painless, quick, and easy, ensuring convenience for scheduling a suitable birth date. These participants believe operative pain only continues in a short time period and harmless compared with labor pain which is long-lasting, and weary procedures accompanied with uncertain risk for mother and baby. Therefore, some women expressed the less-tiring process of CS would bring a quicker recovery in the postpartum period. In addition, the *belief that a child born by CS becomes more beautiful and intelligent* was also shared by some pregnant women; they believe that CS is ideal because their child’s head would not be damaged or deformed through the narrow birth canal or by forceps or vacuum was common among a few of the participants.


*‘I think I should do CS birth. Why I have to bear the pain like my friend? Moreover, I know my pain tolerance, so I think I should do CS birth (Pregnant woman)’.*



*“Yes, I want to have it planned. If everything is possible, and if it suits my health condition, I want to have a scheduled operation. I really don’t want to experience the contraction and the CS operation. It will be like, double the pain. And I don’t like that (Pregnant woman).”*



*“I think, people who are afraid of birth, they will not end up in an unplanned situation. They will check and prepare everything carefully. And they will make sure everything goes as planned and safe. And because they plan carefully they will be less scared (Midwife).”*


Secondly, community and institutional aspects were as follows; most pregnant women from the private interview group and the health care professionals have both mentioned that CS is sometimes requested not only for a woman and her family’s schedule but also because of traditional fortune-telling called ‘phong thuy’ (*Traditional fortune-telling that decides an auspicious day and time*). A child born on a good day and time according to oriental astrology is believed to harmonize with his/her family more in peace, and bring more fortune and prosperity to family members. Even, one of the medical doctors noted some families were trying to avoid giving birth during the period between the western new year (January 1st) and the Vietnamese lunar new year. 


*“Baby born on the right date and time helps the business of the parents to grow. And parent can take good care of the baby (Pregnant woman).”*


A high household income, living in urban areas, attending a private hospital and/or a public tertiary hospital were identified as other factors for the increase of the CS rate the from narratives of the health care professionals. Six health care professionals reported that a child was regarded more as ‘precious’ in the modern era under the one-or-two child policy, and wealthier families are more likely to prefer CS. One pregnant mother and four health care professionals mentioned that CS on-demand without medical needs is more common in private facilities whereas a public tertiary hospital accepts high-risk women who medically indicate CS. 

#### 3.1.2. Theme 2: Mental Strain of Obstetricians

One category and seven subcategories were determined. As for the psychological factor by health care professionals, the *dilemma of obstetricians pressured by family* has emerged; all health care professionals have mentioned they have a strong pressure because they were always monitored by family members, hence it was often the case that CS is decided at an early stage due to their fear of litigation or being targeted on social media. In addition, some obstetricians and midwives have insisted that there is a gray area, which is ambiguous or unclear about the clinical or non-clinical indication of CS in the course of natural labor even if there is a medical guideline, therefore obstetricians’ decision-making may vary depending on each doctor. Furthermore, some health care professionals reported there were some disagreements between obstetricians, who recommend vaginal delivery, and family members, who strongly request a CS without knowing the advantages and disadvantages of CS during labor. It is even the case that some of the family members of pregnant mothers threatened health care professionals for asking about CS. One obstetrician has mentioned it is necessary to obtain agreement about mode of birth not only with the mother but also with other family members during labor. 


*“if indication for forceps or vacuum, I’m willing to do it but when I explain to family members about some procedures risks, family doesn’t want to take the risk. What we can do in case no cooperation? Litigation risk for us, obstetrics is already dangerous. Especially when they have para-clinical diagnosis from outside, already showing brain doppler, diastolic Doppler, or abnormal ultrasound. We check again, normal, but family insists abnormal (Obstetrician)”*



*“I see there’s also the pressure from the family of the parturient woman. They want to have a Caesarean because they’ve already looked for a good date and time. The Doctors have explained to them (that CS without medical reason is not allowed), but they keep on requesting CS (Midwife).”*


#### 3.1.3. Theme 3: Decision Making Process

Four categories and twelve subcategories were identified. More than a half of the women of the private interview group claimed choosing CS is part of a woman and her family’s right (*a woman and family’s autonomy to decide CS*), whereas one woman of the group interview and two health care professionals considered that the choice of having a CS is clearly a medical judgment that should be decided by obstetricians (*CS decision should be made by obstetricians*). 


*“There are many people choose CS on demand (Pregnant woman).”*



*“I have asked. Doctor said, do vaginal birth or have a Caesarean was doctor’s decision, it was not a request from patient (Pregnant woman).”*


Some pregnant women of both private and group interviews admitted that the *media has a strong impact both in positive and negative ways*; Although these pregnant women have recognized that searching for birth-related information through the Internet using their smartphone is quite efficient, they are often negatively affected by videos related to childbirth posted on websites or seen on TV, leading to increased fear towards vaginal birth. Some pregnant women in the private interview admitted they went through great effort to learn about the advantages and disadvantages of their chosen mode of delivery by searching websites and listening to other’s experiences. However, one pregnant woman has concluded how a woman perceives a mode of delivery is always subjective, even if they received broad information through the Internet, therefore, perceived information is always biased, influenced by the individual’s preferences. In addition, listening to relatives and/or friends’ stories, which often described CS as being ‘painless,’ and ‘comfortable,’ affecting the positive perception of CS; one woman decided to request a CS because their relatives recommended CS. However, on the other hand, some of the pregnant women from the group interview have decided to try vaginal delivery encouraged by their relatives and friends.


*“I also watched a few childbirth process video and was so scared after watching the vaginal delivery (Pregnant woman)”*



*“Some mothers had a positive birth experience by CS and they said they did not feel pain at all. their story influenced on strong preference for CS among other mothers (Pregnant woman)”*


### 3.2. Medical Factors

As for medical factors (Table 3), two themes were depicted; (1) maternal risk and (2) institutional factor.

#### 3.2.1. Theme 1: Maternal Risk

Four categories and seven subcategories emerged. A few women and some obstetricians mentioned there is a current trend of over-nutrition and lack of exercise among Vietnamese mothers causing obstructed labor. An older age when giving birth, an increased number of previous CS, and previous abortions were also mentioned among a few health care professionals as risks. One obstetrician mentioned vaginal birth after Caesarean section (VBAC) was not preferred by most of the Vietnamese mothers, therefore, it was rarely conducted. 

#### 3.2.2. Theme 2: Institutional Factor

Three categories and four subcategories were found. As for institutional factors, one medical doctor mentioned a high-risk woman is more likely to be detected because of frequent antenatal screening and ultrasonography (*Influence of antenatal screening*). On the other hand, another obstetrician mentioned that there are still some pregnant mothers who did not receive antenatal checkups appropriately, therefore, severe complication was suddenly identified when those mothers came to the hospital to deliver their baby (*Missed severe pregnancy complications*). Another obstetrician also mentioned that CS is chosen more, as opposed to forceps or vacuum delivery because operative care is much improved in the modern medical environment (*Improvement of operative care*).

## 4. Discussion

Our qualitative study supports the fact that there is a problem of unreasonable CS without clear medical indication, attributed by maternal or family’s request in the urban city of Vietnam. A woman’s fear of childbirth, concern about older age, and lack of self-efficacy for giving childbirth would interact with each other, creating a drive to request CS. Requesting CS may be induced by the belief that CS is superior to vaginal childbirth, spiritual belief for determining the auspicious date of birth, improved economic status, and an increased number of private hospitals. An unreasonable CS being conducted would be also due to psychological fear of litigation and ambiguous clinical indications for CS among obstetricians. In addition, further quantitative investigations on medical factors such as lack of exercise and over-nutrition, previous abortion, previous CS, improvement of antenatal screening suggested in our qualitative study will be important. 

### 4.1. Social Factors for Increased CS

Most of these social factors identified in our study were commonly observed across multi-countries worldwide: fear of childbirth [14,22,23,24,25,26,27,28], lack of self-efficacy for childbirth [22,29,30], and belief that CS is superior to vaginal delivery [10,14,31,32,33], wealthier family [7,13], and private hospital [14,34]. Additionally, determinants of decision making such as a belief that CS is a part of a woman’s autonomy [10,14,35], the strong impact by media [19,36], and/or family and friends [14,19,22,29,37] are consistent with previous studies. On the other hand, the belief in traditional fortune-telling prescribing an auspicious day and time was identified in our qualitative study specific to the Asian region.

Severe fear of childbirth (Tokophobia), which is reported to occur in around 14% of all pregnant women [38], has been recognized as a major contributor to requesting a CS in many previous studies [39]. Three of the primiparous women who participated in a private interview in our study expressed fear of childbirth, which started by hearing other people’s horrible birth stories or information from the Internet resulted in lowering their confidence in giving vaginal birth. Fenwick et al. (2015) described fear brought by other’s stories as ‘imprinted stories’ which become dominant concerns towards their own birth and the complicated birth of siblings or the biological mother provided a strong reference for birth fear [22]. Another case is that the fear of childbirth based on previous traumatic birth was expressed by the multiparous women in our study. Sometimes, there is a misconception that fear of childbirth does not merely indicate fear of labor pain, but a variety of subjects a woman is afraid of, and health care professionals should be aware of this fact. However, it is unlikely to consider that a woman requests CS without medical indication merely because she has a fear of childbirth. Some other social factors may interact with each other elaborately, creating an environment for her to take action more easily. The theory of planned behavior (TPB) may be helpful to understand more about the mechanism. This model presents a theoretical framework consisting of (a) attitude, (b) social normative perceptions, and (c) perceived control over the performance of health behavior, which would explain the likelihood of performing the behavior [40]. First, (a) attitude refers to ‘overall evaluation’ of the behavior considering the possible benefits and the outcomes. Second, (b) social normative perceptions are reflected by the belief of whether others surrounding the individual in his/her community approve or disapprove of the behavior. Third, (c) perceived control over performance is determined by the presence or absence of the facilitators and barriers to behavioral performance. Request for CS can be motivated or justified in the following conditions; belief that CS birth is safe, more comfortable and has a social advantage that is convenient for scheduling CS (*positive attitude*), having an experience of CS being recommended by close friend or family (*social norm*), recognition that CS as part of women’s choice and CS on-demand is often accepted in private setting (*perceived control over performance*).

The belief that traditional fortune-telling decides an auspicious day and time is also consistent with the previous study in China [14]. In our study, most of the participants have mentioned that this cultural belief gives a great impact on requesting CS. The Vietnamese term ‘phong thủy,’ (風水) originating from the Chinese ‘feng shui’, which uses energy forces to harmonize individuals with their surrounding environment; this belief influences not only on deciding the birth date, but also lifestyle. Because the ‘feng shui’ energy element assigned to the year and the date in which an individual was born has an influence over him/her and should be considered along with her zodiac sign and the elements such as water, fire, wind and earth, many parents believe the date of birth of their child would determine whether if the child can harmonize with the parents and bring prosperity of the family. Although ‘phong thủy’ has existed traditionally, the cultural habit to request CS depending on auspicious date and time would be a recent trend along with the improvement of the medical operation. In addition, after the enactment of the Two-Child Policy in the late 1980s, the born child is regarded as more ‘precious.’ Hence, parental feelings of expectation and protection mixed with traditional cultural beliefs might enhance parental preference for scheduling a CS.

Another point is the dilemma of obstetricians. The key message of our medical participants was that obstetricians from public hospitals have tried to follow medical guidelines, however, their effort is sometimes obstructed by strong family pressure demanding a CS birth. In addition, ambiguity in clinical reasoning to perform a CS leads to a variety of clinical judgments among obstetricians. Fear of litigation and ambiguity in clinical reasoning to perform a CS are commonly seen in several previous studies [41]. Not only family members but also obstetricians that are afraid of *unpredictable* complexity in the labor course of vaginal delivery [41]. However, the difference between the obstetricians’ view in our study and those of other previous studies is about a woman’s autonomy to choose a CS. Our obstetricians in the public hospital claimed that performing CS should be based on medical guidelines decided by obstetricians, however, agreement with a woman and her family members is crucial. On the other hand, according to previous studies in Australia and United States, obstetricians regarded CS decision as part of a woman’s autonomy, which should be respected as well as the obstetrician’s decision making [36]. Therefore, our finding implies a woman’s autonomy to decide CS may not be a standard concept in the birth culture of Vietnam, however, family power is strongly affecting on decision making of CS by obstetricians.

### 4.2. Medical Factors for CS

Lack of exercise and over-nutrition, previous CS, improvement of antenatal screening, and previous abortion were identified as medical factors for CS.

As for weight gain during pregnancy, we could not find evidence regarding the trend of the excessive weight gain rate among Vietnamese mothers. A Body Mass Index over 23 Kg/m^2^ is regarded as overweight for the Asian population, which is different from the Western standard [42]. Young et al. [43] identified that 5.9% of the mothers (1439) were overweight before pregnancy, were associated with the heavy weight of their child. Being overweight before pregnancy or weight gain during pregnancy increases the risk of pregnancy/birth complications, leading to a greater need for CS [7,44,45]. Therefore, further epidemiological study investigating the trend of weight gain before and during pregnancy is required. Second, previous CS is another medical indication for CS. Vaginal childbirth after CS (VBAC) is rarely conducted because few women wish to try VBAC, or lack of manpower, despite there being no medical guideline that bans VBAC. Performing a CS birth is already a standard option in the current Vietnamese birth culture. Thirdly, the introduction of ultrasonography contributes to detecting serious risks during pregnancy, however, because minor risk can be detected by ultrasonography, which sometimes increases the anxiety of parents about their child’s health, leading to opting for CS [46]. Because these factors were suggested qualitatively, caution should be taken for representativeness. Hence, further epidemiological study may be required to confirm whether if these factors are related to the rising rate of CS quantitatively.

### 4.3. Strength and Limitations

This study firstly explored factors related to the recent increase of the CS rate qualitatively in an urban city of Vietnam. We included both pregnant mothers who wished and who did not wish to have CS, and health care professionals from a public hospital in our qualitative study, therefore, a variety of factors from these participants were identified in this study.

However, some of the subcategories were stated by only one or two of the participants, hence, caution should be paid for the representativeness. In addition, there is a limitation that we did not include obstetricians working in a private clinic or hospital although the private hospital was identified as a major factor in many previous studies. The participating health care professionals were from public hospitals, qualitative interviews, including from private settings, may yield different results. However, it would be difficult to interview health care professionals in private hospitals and ask them to speak up regarding the situation.

### 4.4. Clinical Implications

First, the issue of unnecessary CS without clear medical indication should be understood at the community level. Because the rising rate of CS requests is not only a matter of individual (family) decision making caused by fear, or/and cultural belief such as fortune-telling, but is also the social phenomenon which is increasing in the whole community. Some beliefs identified from this study such as ‘CS is safe and more comfortable’ and ‘requesting CS is more acceptable in society’ cannot be generalized in the society, however, these viewpoints may become a common idea in the community. Providing appropriate information regarding the procedure and the possible risks of CS as well as the natural process of vaginal delivery in public is important.

Second, a recent Cochran review concluded the evidence regarding the effect of nurse-led education and childbirth preparation intervention may be effective [47]. This qualitative study supports the fact that there are some women who are suffering from fear of vaginal childbirth, who are opting for CS, thus, these women should be mentally supported by midwives. Emotional encouragement and building trust between health care professionals and a mother are reported to be effective to motivate a woman to try a vaginal delivery. In addition, involving a partner and other family members is especially important in the birth context of Vietnam.

Third, the Cochran review also indicates guidelines with a mandatory second opinion and post-Caesarean surveillance can lead to a small reduction in CS [47]. Continuous monitoring of hospitals by the government may be necessary. Furthermore, a multi-faced approach targeting not only women and family, but also an institutional level, such as a mandatory reporting system, and implementing clear clinical guidelines, and community approaches will be necessary.

Finally, our study identified obstetricians’ psychological stress was severe. Manpower is one of the most important components which maintains a safe birth environment. Encouraging the existing birth culture that obstetricians try to build a good relationship with family members and education targeted towards family members is also necessary for enhancing their compliance.

## 5. Conclusions

The paper provides evidence of unnecessary Caesarean sections without clear medical indication, which was requested by women and family members in an urban area of Vietnam. Requesting a CS can be induced by a woman and family’s psychological fear of childbirth, concern about older age, and lack of self-efficacy for giving childbirth, belief that CS is superior to vaginal childbirth, and spiritual belief for determining the auspicious date of birth. In addition, psychological fear of litigation and ambiguous clinical indications for CS among obstetricians are found to be other social determinants.

## Figures and Tables

**Table 1 healthcare-08-00041-t001:** Characteristics of participants.

Pregnant Women (N = 29)	N	%	Health Care Professionals (N=19)	N	%
**Birth parity**			**Profession**		
0	17	59%	Obstetricians	4	21%
1	9	31%	Midwife	15	79%
2	3	10%	**Gender**		
**Age**			Male	2	11%
20–24	4	14%	Female	17	89%
25–29	10	34%	**Age**		
30–34	8	28%	25–29	5	26%
35–39	7	24%	30–34	0	0%
**Occupation**			35–39	10	53%
Full time employee	21	72%	40–44	4	21%
Part time employee	0	0%	45–	0	0%
Own business	2	7%	**Years in post**		
House wife	6	21%	1–4	3	16%
**Pregnancy complexity**			5–9	2	11%
Normal	28	97%	10–14	5	26%
Abnormal	1	3%	15–19	4	21%
**Gestational weeks**			20–	5	26%
Second trimester	15	52%			
Third trimester	14	48%			
**Desire for CS**					
Yes	10	34%			
No	19	66%			

**Table 2 healthcare-08-00041-t002:** Social factors for increased CS.

Category (Subcategory)		G1 N	G2 N	G3 N	Category (Subcategory)		G1 N	G2 N	G3 N
**Theme 1: Request for CS**					**Theme 1: Request for CS (continued)**				
1. Fear of vaginal childbirth					5. Belief CS is safe and more comfortable				
1.1.	Labor pain	10	7	3	5.1.	CS can avoid any possible risk of vaginal delivery	10	4	
1.2.	Unexpected risk for mother and baby	7	1	7	5.2.	CS is painless than vaginal delivery	4		
1.3.	Episiotomy	6		1	5.3.	CS surgery is quick and easy	3	3	
1.4.	Suffering for a long period of time	5	2		5.4.	CS enables quick physical recovery	3		
1.5.	Negative experience of previous childbirth	3			5.5.	Decision of CS makes mother feel secured	2		
1.6.	Injection	3			6. Belief child born by CS becomes beautiful and intelligent				
1.7.	Losing control and having a panicky attitude	2			6.1.	Child born by CS is more intelligent than other	2	1	
1.8.	Negative attitude towards vacuum or forceps delivery	1		3	6.2.	Child born by CS is more beautiful in shape	1	1	
2. Lack of self-efficacy for childbirth					7. Traditional fortune-telling decide auspicious day and time				
2.1.	Lack of self-confidence with giving birth	3	2		7.1.	Belief that baby born in auspicious day and time brings fortune for baby and family	10	7	14
2.2.	Self-recognition of having a low tolerance with pain	3			7.2.	Wish to avoid giving birth during the period between Western new year holiday and traditional new year holiday			1
3. Concern about older age					8. Preference of receiving CS birth among wealthier family in urban areas.				
3.1.	Concern about increased risk of complexity	4			8.1.	Child is more regarded as ‘precious’			6
3.2.	Older age is a valid reason for choosing CS	2			8.2.	CS is more preferred among wealthier family			2
3.3.	Lack of self-confidence with her strength due to her older age	2			8.3.	CS is more preferred among family living in urban areas.			2
					8.4.	CS birth is becoming economically affordable			1
4. Belief CS has social advantage					9. CS varies in hospitals and regions				
4.1.	CS has advantage of scheduling birth date	6	3		9.1.	CS on request is conducted in private settings	1		4
4.2.	CS gives less damage on sexuality	1	1		9.2.	CS rate of urban tertiary hospitals is higher than other district hospitals among high risk mothers.			1
4.3.	CS has advantage for receiving sterilization.			1	9.3.	Misunderstanding that a provincial hospital is ready for CS among some family members.			1
**Theme 2: Mental strain of obstetricians**					**Theme 3: Decision-making process**				
10. Dilemma of obstetricians pressured by family					11. A woman and family’s autonomy to decide CS				
10.1.	Obstetricians sometimes decide CS earlier in order to ensure safety and protect themselves from being accused			9	11.1.	Belief CS is a choice of a mother and her family	5		
10.2.	Family pressure on obstetricians to suggest CS			7	11.2.	Belief request CS is more acceptable in society	8		
10.3.	Gray area to judge situations as normal or abnormal in labor childbirth			4	12. CS should be decided by obstetricians				
10.4.	Lack of knowledge about CS among mothers and families			4	12.1.	Disagreement between family and obstetricians for choosing CS.	2		
10.5.	Informed consent to avoid a conflict between family and obstetricians			3	12.2.	CS decision should be made following guideline by obstetricians		1	2
10.6.	Fear of being claimed by family through social media			2	13. Media has strong impact				
10.7.	Agreement of family is necessary in decision making of CS			1	13.1.	Internet as an efficient tool to collect information	6	3	
					13.2.	Fear of vaginal childbirth brought by video on the Internet and TV	3	3	
					13.3.	Biased information on the Internet	1		
					13.4.	Internet is accessible by everyone, however people get different conclusions		3	
					13.5.	Great effort to capture maximum information through a variety of resources.	3		
					14. Strongly influenced by family and friends.				
					14.1.	Feeling of confusion listening to opposite perceptions for mode of delivery	4		
					14.2.	Decision making of CS influenced by relatives and friends	1	7	
					14.3.	Fear of vaginal childbirth brought by other’s story	1		

G 1: pregnant women who wished CS (N = 10), G 2: pregnant women who did not wish CS (N = 19), G 3: healthcare professionals (N = 19), N: Number of participants mentioned about subcategory.

**Table 3 healthcare-08-00041-t003:** Medical factor for CS.

Category (Subcategory)			G1 N	G2 N	G3 N
**Theme 1: Maternal risk**					
1. Over-nutrition and lack of exercise					
	1.1.	Lack of exercise and strength to give birth		2	1
	1.2.	Lack of exercise and overweight			5
2. Older maternal age of giving birth					
	2.1.	A trend of giving birth at an older age would increase medical indications of CS			2
3. Previous CS					
	3.1.	Previous CS is a valid medical reason for choosing CS	1	1	2
	3.2.	Shortage of obstetricians who assist vaginal birth after Caesarean (VBAC)			1
	3.3.	Few mothers prefer to try VBAC because of its risks			1
4. Increased previous abortion					
	4.1.	Increased number of abortion leads anomaly of placenta			1
**Theme 2: Institutional factor**					
5. Influence of antenatal screening					
	5.1.	Improvement of antenatal screening detects more high-risk women indicated for CS			1
	5.2.	Introduction of ultrasonography detects more abnormality			1
6. Missed severe pregnancy complications					
	6.1.	Pregnancy complications detection missed, due to misunderstanding and lack of knowledge among mothers and family			1
7. Improvement of operative care					
	7.1.	Improvement of operative care replaced instrumental delivery (forceps or vacuum) to CS in modern Vietnam.			1

G 1: pregnant women who wished CS (N = 10), G 2: pregnant women who did not wish CS (N = 19), G 3: healthcare professionals (N = 19), N: Number of participants mentioned about subcategory.

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
