# Peer review of "Reasons for Increased Caesarean Section Rate in Vietnam: A Qualitative Study among Vietnamese Mothers and Health Care Professionals"

_healthcare, 2020, doi:10.3390/healthcare8010041_

Round 1
Reviewer 1 Report
Title: might be changed from the interrogation sentence to descriptive. Abstract: The CS rate in Vietnam is higher than the recommended rate by the WHO. Mention the rates in both cases. What is the rate of CS in Vietnam? What is the recommended rate of CS by WHO? How many times higher the CS rate from the recommended rate of WHO in Vietnam? Methodology: Study period: not mentioned anywhere. Who conducted the study (name of the organization/s), not mentioned anywhere? What was the justification of excluding <20 years of aged pregnant women from the study? Nothing is mentioned about Ethical Considerations of the study. Results: Tables are too long and difficult to understand, esp. Table 2a. Findings can be presented in different ways so those can be easily understandable. Unnecessary CS is an important area to discuss. There might be some recommendations derived from the study findings and can be presented as Recommendations.
Author Response
Respond to reviewer’s comments
Reviewer 1.
Thank you for giving me the opportunity to submit a revised draft of my manuscript titled ‘Why Caesarean section rate is increasing in Vietnam: Voice of Vietnamese mothers and health care professionals’ to Health care (MDPI). We are grateful to the reviewers for their insightful comments on my paper. We have been able to incorporate changes to reflect most of the suggestions provided by the reviewers.
Here is a point-by-point response to the reviewers’ comments and concerns.
|
1. Title Title might be changed from the interrogation sentence to descriptive. |
|
We modified the title accordingly, (Title) Reasons for increased Caesarean section rate in Vietnam: A qualitative study among Vietnamese mothers and health care professionals |
|
2. Abstract The CS rate in Vietnam is higher than the recommended rate by the WHO. Mention the rates in both cases. What is the rate of CS in Vietnam? What is the recommended rate of CS by WHO? |
|
The following is corrected so that the sentence is more clear. (Abstract) Caesarean section rate in urban Vietnam is 43% in 2014, which is more than twice as high as the recommended rate (10 -15 %) by World Health Organization. |
|
3. Methodology Study period: not mentioned anywhere. Who conducted the study (name of the organization/s) |
|
The following is added on Method (Procedure). (P3, Line110)This study was conducted between April and July in 2018. With regard to organizers, because it is a collaborative study, multiple stakeholders collaborated as mentioned in study protocol. 1) Recruitment of mothers was done by medical staffs in hospitals and recruitment of medical staffs was done by government officers in Khanh Hoa Health Service. 2) Interviews were conducted by trained two Vietnamese obstetricians and a midwife. |
|
4. Methodology What was the justification of excluding <20 years of aged pregnant women from the study? Nothing is mentioned about Ethical Considerations of the study. |
|
Thank you very much for valuable comments. Although some study regard any person who is over 18 years old as ethically competent, we used the cut-off point of 20 years old because of WHO definition of adolescent.
The sentences are added as follows, (P3, Line 96) Adolescent is defined as a person aged between 10 and 19 according to WHO [20], who is not fully matured, has three ethical issues; consent, confidentiality, and decision making[21]. Although many previous studies regard age of 18 as being capable of these ethical issues, the authors decided to exclude any person who is under the age of 20 years following WHO definition. |
|
5. Results Tables are too long and difficult to understand, esp. Table 2a. Findings can be presented in different ways so those can be easily understandable. |
|
Thank you for the valuable comment. We have formatted tables. Hope these tables clear to readers. |
|
6. Unnecessary CS is an important area to discuss. There might be some recommendations derived from the study findings and can be presented as Recommendations. |
|
The following sentences were added in clinical implementation part,
(P14, Line 416) First, the issue of unnecessary CS without clear medical indication should be aware at community level. Because requesting CS is not only a matter of individual (family) decision making caused by fear, or/and cultural belief such as fortune telling, but also is the social phenomenon which is rising in the whole community. Some believes identified from this study such as ‘CS is safe and more comfortable’ and ‘requesting CS is more acceptable in society’ cannot be generalized in the society, however these viewpoints may become a common idea in the community. Providing appropriate information including the procedure and the possible risk of CS as well as natural process of vaginal delivery in public is important. |
Thank you very much.
Reviewer 2 Report
Thank you for the opportunity to review your manuscript.
First of all, the manuscript would benefit of proofreading, some sentences are very long, verbs are not conjugated, there is confusion due to plural and singular, ...
P.3, line 107: each session lasted from ...: I suggest to place this under the results section, not in the procedure section
Who was selected to take part of the interview is not clear in the text, it became clear for me in the results section.
There are exclusion criteria, but in table 1, there is 1 respondent with an abnormal pregnancy. Should she be excluded?
Table 2 a and 2 b: I have a problem with the way the qualitative research is performed. The authors tried to quantify their qualitative results, by counting how many times (sub)categories are mentioned. This is not the way to perform this kind of research.
In Table 2 b the (sub)categories even don't match the number in the text (p. 5 line 261 and line 269): given the information in the table, I count 4 categories and 7 subcategories in theme 1, 3 categories and 4 subcategories in theme 2, p. 12).
P. 14: what do the authors call 'overweight'? A BMI of 23 is normal in the Western society (normal is between 19 and 25). Is this different in Vietnam? Then this should be clarified.
The authors performed focus groups with health professionals, but I only read results about the obstetricians. What about the midwives (15/19 health care professionals)? How do they thing about this topic? Do they feel the same as the obstetricians, do they experience also pressure of the women and their family?
P. 16: in the interview guide ideal care is mentioned. Was this topic discussed with the women or in the focus groups? What do women and health care workers prefer as ideal care?
Author Response
Respond to reviewer’s comments
Reviewer 2
Thank you for giving me the opportunity to submit a revised draft of my manuscript titled ‘Why Caesarean section rate is increasing in Vietnam: Voice of Vietnamese mothers and health care professionals’ to Health care (MDPI). We are grateful to the reviewers for their insightful comments on my paper. We have been able to incorporate changes to reflect most of the suggestions provided by the reviewers.
Here is a point-by-point response to the reviewers’ comments and concerns.
|
1. English proofreading |
|
Thank you very much for your comment. We finished professional proofreading on the revised manuscript. Hope this revised version understandable. |
|
2. P.3, line 107: each session lasted from ...: I suggest to place this under the results section, not in the procedure section Who was selected to take part of the interview is not clear in the text, it became clear for me in the results section. There are exclusion criteria, but in table 1, there is 1 respondent with an abnormal pregnancy. Should she be excluded? |
|
I moved the sentence (P3, line 107) to the result part. Although any person who had serious pregnancy complication was excluded in this study, one pregnant woman had minor pregnancy complication. We modified sentence as below; (P4, Line 142) Most of participants had no serious pregnancy complexity but one participant had minor pregnancy complication(anemia). |
|
3. Table 2 a and 2 b: I have a problem with the way the qualitative research is performed. The authors tried to quantify their qualitative results, by counting how many times (sub)categories are mentioned. This is not the way to perform this kind of research. |
|
Thank you very much for letting us know. We missed to explain about the content analyses we followed. Following is added sentences, (P3. Line 103) With regard to analysis, a content analysis was adopted. Content analysis is often used as a combination of qualitative and quantitative methods. The qualitative way draws the presence, meanings and relationships of such certain words, themes, or concepts within some qualitative data. The quantitative way focuses on counting the occurrence of certain words and phrases in different context [22]. |
|
4. Table 2 b the (sub)categories even don't match the number in the text (p. 5 line 261 and line 269): given the information in the table, I count 4 categories and 7 subcategories in theme 1, 3 categories and 4 subcategories in theme 2, p. 12). |
|
We corrected accordingly. |
|
5. P. 14: what do the authors call 'overweight'? A BMI of 23 is normal in the Western society (normal is between 19 and 25). Is this different in Vietnam? Then this should be clarified. |
|
Thank you for the valuable comment. There is a different cut-off points recommended for Asian population (Lancet, 2004). The following sentence is added so that it is more understandable. (P13, Line 384) Body Mass Index over 23 Kg/m2 is regarded as overweight for Asian population, which is different from western standard[44]. Young et al [45] identified that 5.9% of total 1439 mothers was overweight before pregnancy, was associated with heavy weight of child. Over-weight before pregnancy or over weight gain during pregnancy increase a risk of pregnancy/birth complications leading to a greater need for CS [7] [46] [47]. |
|
6. The authors performed focus groups with health professionals, but I only read results about the obstetricians. What about the midwives (15/19 health care professionals)? How do they thing about this topic? Do they feel the same as the obstetricians, do they experience also pressure of the women and their family? |
|
Thank you very much for your comments. Actually, because there was no big difference between obstetrician and midwives during interviews. Midwives were more likely to mention about a mother’s psychological fear as a reason for requesting CS than doctors. Therefore, we combined the two professions into one group of health care professionals in the results. However, we added some more narratives expressed by midwives.
(P5, Line 194) “I think, people who are afraid of birth, they will not end up in an unplanned situation. They will check and prepare everything carefully. And they will make sure everything goes as planned and safe. And because they plan carefully they will be less scared (Midwife).”
(P6, Line240) “I see there’s also the pressure from the family of the parturient woman. They want to have a caesarean because they’ve already looked for a good date and time. The Doctors have explained to them (that CS without medical reason is not allowed), but they keep on requesting CS(Midwife).” |
|
7. P. 16: in the interview guide ideal care is mentioned. Was this topic discussed with the women or in the focus groups? What do women and health care workers prefer as ideal care? |
|
Thank you for asking about it. We asked the question during the interview, but their answers were quite Broad and too much to summarize in this paper. It was not only focusing on Caesarean section but also other general topics such as nutrition etc. Although we are considering to summarize the findings in another paper, we decided not to mention about it this time. |
Thank you very much.
Round 2
Reviewer 2 Report
I read the revised version and I agree that this version is ready for publication. One Remarque: there are some track changes visible in table 2 b.